# *Exploratory study of using Magnetic resonance Prognostic Imaging markers for Radiotherapy In Cervix cancer (EMPIRIC): a prospective cohort study protocol*

Mohammed Abdul-Latif [1,2] Amani Chowdhury,[1,2] Hannah Tharmalingam,[2] N Jane Taylor [3] Amish Lakhani,[3] Anwar Padhani,[3] Peter Hoskin,[1,2] Yatman Tsang[2,4]

¹Division of Cancer Sciences, The University of Manchester, Manchester, UK
²Clinical Oncology, Mount Vernon Cancer Centre, Northwood, UK
³Paul Strickland Scanner Centre, Northwood, UK
⁴Radiation Medicine, Princess Margaret Hospital Cancer Centre, Toronto, Ontario, Canada

**Correspondence to**
Dr Mohammed Abdul-Latif;
m.abdul-latif@nhs.net

## ABSTRACT

**Introduction** Radical chemoradiotherapy represents the gold standard for locally advanced cervical cancer. However, despite significant progress in improving local tumour control, distant relapse continues to impact overall survival. The development of predictive and prognostic biomarkers is consequently important to risk-stratify patients and identify populations at higher risk of poorer treatment response and survival outcomes. Exploratory study of using Magnetic resonance Prognostic Imaging markers for Radiotherapy In Cervix cancer (EMPIRIC) is a prospective exploratory cohort study, which aims to investigate the role of multiparametric functional MRI (fMRI) using diffusion-weighed imaging (DWI), dynamic contrast-enhanced (DCE) and blood oxygen level-dependent imaging (BOLD) MRI to assess treatment response and predict outcomes in patients undergoing radical chemoradiotherapy for cervical cancer.

**Methods and analysis** The study aims to recruit 40 patients across a single-centre over 2 years. Patients undergo multiparametric fMRI (DWI, DCE and BOLD-MRI) at three time points: before, during and at the completion of external beam radiotherapy. Tissue and liquid biopsies are collected at diagnosis and post-treatment to identify potential biomarker correlates against fMRI. The primary outcome is to evaluate sensitivity and specificity of quantitative parameters derived from fMRI as predictors of progression-free survival at 2 years following radical chemoradiotherapy for cervical cancer. The secondary outcome is to investigate the roles of fMRI as predictors of overall survival at 2 years and tumour volume reduction across treatment. Statistical analyses using regression models and survival analyses are employed to evaluate the relationships between the derived parameters, treatment response and clinical outcomes.

**Ethics and dissemination** The EMPIRIC study received ethical approval from the NHS Health Research Authority (HRA) on 14 February 2022 (protocol number RD2021-29). Confidentiality and data protection measures are strictly adhered to throughout the study. The findings of this study will be disseminated through peer-reviewed publications and scientific conferences, aiming to contribute to the

## STRENGTHS AND LIMITATIONS OF THIS STUDY

⇒ Novel imaging approach: Multiparametric functional MRI (fMRI) including T2, diffusion-weighed imaging, dynamic contrast-enhanced and blood oxygen level-dependent-MRI.
⇒ Temporal data: fMRI obtained prior, during and at the end of chemoradiotherapy will capture longitudinal changes in tumour and fMRI-derived parameters.
⇒ Parallel molecular analysis: Tissue and liquid biopsies (circulating tumour cells and circulating tumour DNA) will be collected and independently analysed to assess hypoxic and tumour burden respectively and correlated with fMRI sequences.
⇒ Integrated design: fMRI and molecular samples are collected during chemoradiotherapy with limited additional appointments or effort required for enrolled patients, allowing for smooth translation of imaging and biopsy protocol into wider clinical practice.
⇒ Single-site study: Patient recruitment is limited to eligible patients referred for chemoradiotherapy at a single cancer centre.

growing body of evidence on the use of multiparametric MRI in cervical cancer management.
**Trial registration number** NCT05532930.

## INTRODUCTION

Cervical cancer is the fourth most common cancer affecting women worldwide.[1] While surgery is suitable for early-stage disease, advanced stage presentations require a combination of chemoradiotherapy (CRT) and image-guided brachytherapy boost as the standard of care. Although local control rates are promising (92% at 5 years), the impact of metastatic disease on overall survival highlights the need to identify high-risk populations with poorer treatment outcomes.[2] To

address this, predictive and prognostic biomarkers are needed.

The use of predictive biomarkers derived from imaging can potentially guide treatment decisions and improve outcomes. Imaging biomarkers are advantageous as they are non-invasive, allow for multiple assessments and provide a comprehensive evaluation of a tumour in its entirety. Unlike biopsies, imaging biomarkers provide valuable spatiotemporal data, which can capture evolving intratumour heterogeneity to better predict tumour control probability.[3]

MRI holds promise as a source of imaging biomarkers for cervical cancer. Traditional T1-weighted and T2-weighted anatomical MRI techniques are commonly used for staging, diagnosing and monitoring cervical cancer.[4] Additionally, advanced MRI techniques, known as functional MRI (fMRI), can provide valuable biological data. These techniques include diffusion-weighted imaging (DWI), which measures cellular density using water motion assessment; dynamic contrast-enhanced imaging (DCE), which assesses blood flow and vascular permeability; and blood oxygen level-dependent imaging (BOLD), which measures changes in tissue oxygenation. By leveraging these advanced techniques, fMRI may have a role in offering predictive and prognostic biomarkers for cervical cancer.[5]

This has been clarified in a systematic review, which details prospective studies using different fMRI techniques in cervical cancer. DWI and DCE-MRI represent the most used techniques, followed by BOLD-MRI. Furthermore, the majority of studies evaluated a single technique rather than a combination. This has led to a lack of consensus confounded by 'multiple interacting factors, including histology, grade, hypoxia, vascularity and cellularity that can influence fMRI-derived parameters'. To account for this, future studies will need to employ combinations of fMRI techniques to generate 'a more holistic assessment of the tumour, simultaneously assessing multiple tumour features to improve outcome prediction. Refinement of fMRI as a predictive and prognostic marker can be improved further by combining clinicopathological scoring and or genomic signatures for hypoxia.[6]

In addition to a multiparametric approach, fMRI needs to be repeated across treatment to discern tumour response and relate it to fMRI-derived parameters. Spatial data can also be acquired—an example of which DCE-MRI can identify a 'functional risk volume' within the tumour associated with poorer response to treatment. Tumours with a 'functional risk volume' displayed poor vascularity and tumour permeability.[7] More recently, a study demonstrated combining DWI-derived and DCE-derived parameters to describe the 'hypoxic fraction' of cervical tumours linking this to disease-free survival.[8]

The Exploratory study of using Magnetic resonance Prognostic Imaging markers for Radiotherapy In Cervix cancer (EMPIRIC) study aims to build on previous studies and address the gaps in the current evidence base by performing multiparametric fMRI (DWI, DCE and BOLD-MRI) across treatment in patients with cervical cancer, alongside collection of blood samples for circulating tumour cell (CTC) and circulating tumour DNA (CtDNA) and tissue biopsies to facilitate parallel molecular analysis. This holistic approach to tumour characterisation will provide the tools to generate a meaningful predictive and prognostic model for personalised radiotherapy for cervical cancer.

## METHOD

### Study design and setting

EMPIRIC (ClinicalTrials.gov NCT05532930) is a prospective, non-randomised, single-centre (Mount Vernon Cancer Centre, UK) observational study in which patients with locally advanced cervical cancer fit for CRT are included.

The study aims to recruit 40 patients in total over an accrual period of 36 months at Mount Vernon Cancer Centre, UK. The study started in August 2022 and is actively recruiting. It is a single cohort study, where patients undergoing CRT (45 Gy in 25# of external beam radiotherapy with concurrent weekly cisplatin, followed by image-guided brachytherapy boost) for cervical cancer undergo fMRI before, during and at the end of their treatment, along with analysis of CTC, CtDNA and tissue biopsies. Online supplemental table 1 provides an overview of the study schedule.

### Patient and public involvement and engagement

As part of trial development, evaluation and input from lay members of the funding committee were collected. Patients reviewed in outpatient clinics were consulted in addition to a patient advocate. We are continuing to gather patient feedback from enrolled participants and their relatives on the acceptability of added interventions (fMRI, blood tests and biopsies).

### Consent and withdrawal

Written informed consent will be obtained for all patients included in the study before they are registered by a clinician within the EMPIRIC research team.

All patients will be informed of the purpose of the investigation and the possible risks and side effects involved with the extra scans and contrast administration through patient information sheets and at the time of consent. Patients will be informed as to the strict confidentiality of their patient data, but that their medical records, including trial imaging, may be reviewed for study purposes by authorised individuals other than their treating physician. Participants are free to withdraw at any time, or the discretion of the EMPIRIC chief or principal investigator. In the event of withdrawal, any data that have been collected will be kept and potentially included in any analyses.

### Eligibility criteria

The inclusion criteria are as follows:

- ► Histologically confirmed International Federation of Gynecology and Obstetrics (FIGO) stage Ib2–IVa squamous, adeno or adenosquamous carcinoma of the cervix.
- ► Clinically and/or radiographically documented measurable disease with at least one site of disease unidimensionally measurable as per Response Evaluation Criteria in Solid Tumours (RECIST) 1.1
- ► All detectable disease including pelvic/para-aortic nodes encompassable within radical high-dose radiation field.
- ► Deemed suitable and fit for radical CRT.
- ► Eastern Cooperative Oncology Group (ECOG) performance status 0–1.
- ► Aged 18 and over.
- ► Documented negative pregnancy test (if applicable).
- ► Capable of providing written or witnessed informed consent according to International Council for Harmonisation of Technical Requirements for Registration of Pharmaceuticals for Human Use (ICH)/ Good Clinical Practice (GCP) and national/local guidelines prior to registration.

The exclusion criteria are as follows:

- ► Previous pelvic malignancy (regardless of interval since diagnosis).
- ► Previous malignancy not affecting the pelvis (except basal cell carcinoma of the skin) where disease-free interval is less than 10 years.
- ► Evidence of distant metastasis, that is, any non-nodal metastasis beyond the pelvis.
- ► Previous pelvic radiotherapy.
- ► Prior diagnosis of Crohn's disease or ulcerative colitis.
- ► Uncontrolled cardiac disease (defined as cardiac function which would preclude hydration during cisplatin administration).

- ► Previous record of allergic reaction to gadolinium-based contrast media and any other contraindication to MRI.
- ► Any psychological, familial, sociological or geographical condition potentially hampering compliance with the study protocol and follow-up schedule; those conditions should be discussed with the patient before registration in the study.
- ► Participation in any other interventional trials.

## fMRI assessment

Each multiparametric fMRI scan will consist of sequences: DCE-MRI, DWI-MRI, BOLD-MRI, quantitative T1-weighted and T2-weighted MRI and will be carried out at three time points- prior to CRT (RT planning), week 2 of CRT (fraction 10) and week 5 of CRT prior to brachytherapy (fraction 25). Delineations of the gross tumour volume (GTV) using T2-weighted MR sequences for each case will be performed by a clinical oncologist and verified by a consultant radiologist. The GTV tumour volume reduction rate (TVRR) will be determined between these three time points as defined on anatomic sequences. Kinetic modelling will be applied in DCE-MRI images to assess quantitative kinetic parameters in the GTV related to blood volume, blood flow, interstitial volume and tissue permeability. DWI-MRI sequences will be used to calculate ADC maps to provide information related to tumour cellularity and necrosis. Signal changes on BOLD-MRI sequences will be used to calculate the T2 relaxation rate reflecting intravascular deoxyhaemoglobin concentration, and therefore, providing information related to tumour oxygenation. Anatomic T1 and T2 imaging will be used for radiomic feature extraction and textural analysis through quantitative mapping.

The details of each MRI and radiomic parameter are summarised in table 1.

**Table 1** fMRI sequences with corresponding parameters used to assess cervical tumour response during radical chemoradiotherapy

| Imaging | Parameters | Biological correlates |
|---|---|---|
| DCE-MRI | Extravascular extracellular space volume fraction $V_e$ (%) | Blood volume, blood flow, interstitial volume and tissue permeability |
| | Transfer constant $K^{trans}$ (/min) | |
| | Rate Constant $k_{ep}$ (/min) | |
| | Blood volume fraction $v_p$ (%) | |
| DWI-MRI | Apparent diffusion coefficient ADC ($um^2/s$) | Tumour cellularity, architectural complexity and necrosis |
| BOLD-MRI | BOLD-based reversible transverse relaxation rate $R2^*$ (/s) | Intravascular deoxyhaemoglobin concentration |
| T1 and T2 weighted MRI | Relaxation times (/s) | Tissue physical characters affecting water relaxation and anatomical information |
| T2-weighted±DWI MRI (Grey level co-occurrence matrix) | Second-order Haralick textural radiomic features | Intratumoural heterogeneity |

BOLD, blood oxygen-level dependent; DWI, diffusion-weighed imaging; fMRI, functional MRI.

## Translational biological assessment

Tumour tissues from the diagnostic biopsy and a second biopsy taken at the time of brachytherapy will be collected. The unstained biopsy slides will be stained for immunohistochemical markers of hypoxia (CA IX, GLUT-1), vascularity (VEGF) and epithelial mesenchymal transition (CD44, SOX-2). These molecular markers at various time points will be correlated with imaging parameters and evaluated independently and in combination with respect to prognostic impact. Peripheral blood samples will be taken at the three imaging time points (pre-CRT, fraction 10 and fraction 25 of CRT) for analysis of CTCs. CTCs will be isolated from peripheral blood samples via an optimised enrichment technique using CD45 antibody. Based on branched DNA signal amplification technology, CTCs will be classified into three subpopulations with respect to metastatic potential using epithelial mesenchymal transition markers: epithelial CTCs (EpCAM or CK8), mesenchymal CTCs (vimentin or TWIST) and mixed phenotype CTCs (both markers). The CTC phenotype ratio at various time points will be correlated with imaging parameters and evaluated independently and in combination with respect to prognostic impact (progression-free survival and overall survival).

## Radiotherapy-related assessment

Quantitative MR parameters (BOLD signals complimented by perfusion information from DCE-MRI and cellularity information from DWI-MRI) derived at various time points during treatment will be used to create a high-risk map that can be delineated and registered onto the anatomic sequences of the planning MRI performed at the time of image-guided brachytherapy. The map will be used to delineate a residual high-risk tumour volume (HTV-T$_{res}$) within the high-risk clinical target volume (HR-CTV) for brachytherapy. Original treatment plans will then be reoptimised escalating the HTV-T$_{res}$ to 140% of the original HR-CTV prescription (9.8 Gy/fraction) with the aim of achieving the volume of HTV-T$_{res}$ receiving 140 Gy (HTV-T$_{res}$V$_{140}$) $\geq 90\%$ and 98% of HTV-T$_{res}$ to receive over 100 Gy (D$_{98}\geq100$ Gy) while maintaining HR-CTV coverage and organ-at-risk dose-volume histograms within tolerance.

## Follow-up schedule

All patients from this study will be followed up at 3 months, 6 months, 12 months and then yearly as outlined in table 2. Treatment outcome data will be collected for all patients until 2 years post-CRT end date.

Objective tumour response on MR in patients with measurable disease will be assessed by RECIST V.1.1 week 2 of CRT (fraction 10), week 5 of CRT prior to brachytherapy (fraction 25) and 3 months post the end date of CRT.

Progression-free survival (local, nodal, distant) and overall survival at 1 and 2 years will be measured. The study will close once the final patient accrued has completed a 2-year follow-up.

**Table 2** Post-treatment follow-up and imaging schedule as per standard of care

| Months | 1 | 3 | 6 | 12 | 24 |
|---|---|---|---|---|---|
| Clinical assessment* | ✓ | ✓ | ✓ | ✓ | ✓ |
| MR RECIST assessment | | ✓ | | ✓ | ✓ |

✓ Indicates when patient will receive stated treatment or undergo stated investigation.
* All post-treatment imaging follow up procedures (including Positron Emission Tomography (PET)/CT and MRI scanning) are carried out per standard of care at Mount Vernon Cancer Centre. RECIST, Response Evaluation Criteria in Solid Tumours.

## Outcomes

### Primary outcome

1. To evaluate sensitivity and specificity of quantitative parameters derived from DCE-MRI, DWI-MRI and BOLD-MRI as predictors of progression-free (local, nodal, systemic) survival at 2 years post-treatment in patients with locally advanced cervical cancer after CRT.

### Secondary outcomes

1. To evaluate sensitivity and specificity quantitative parameters derived from DCE-MRI, DWI-MRI and BOLD-MRI as predictors of overall survival at 2 years in patients with locally advanced cervical cancer after CRT.
2. To evaluate sensitivity and specificity quantitative parameters derived from DCE-MRI, DWI-MRI and BOLD-MRI as predictors of the TVRR in response to CRT of patients with locally advanced cervical cancer.
3. To apply radiomics for derivation of textural imaging features as predictors of progression-free (local, nodal, systemic), overall survival and TVRR after CRT for cervix cancer.
4. To correlate temporal changes in quantitative MRI parameters during CRT with molecular markers from tissue biopsies and CTCs isolated from peripheral blood samples.
5. To explore the feasibility of using quantitative MRI parameters to provide a roadmap in identifying high-risk regions for potential dose escalation with radiotherapy.

## Sample size and power

In the absence of evidence in published literature on which to derive sample size calculations in the context of an exploratory feasibility study, formal statistical power calculations have not been performed. An aim of exploratory studies is to determine feasibility of specific study aspects, subsequently allowing for estimation of sample sizes in the future studies.

The proposed sample size of 40 patients is a convenience-based sampling reflecting the annual estimate of patients with locally advanced cervical cancer likely to be eligible for this study at Mount Vernon Cancer Centre. From January 2019 to December 2019, there were a total of 30 patients with locally advanced cervical cancer treated with

radical CRT. Assuming a conservative patient decline/withdrawal rate of 30%, the estimated annual number of complete study accruals is 21. This should enable a total number of 40 patients to be recruited over a period of 24 months. In a mixed population of cervix patients, a 2-year recurrence rate (local, nodal and distant) of 35% is assumed resulting in an estimated number of 14 events by the end of the study period.

## Data handling and record keeping
All study data will be securely held and managed in accordance with relevant statutory data protection legislation. Technical appendix, statistical code and dataset will be stored and accessible through a local server. Information about study subjects will be kept confidential and managed according to Trust policies. Patients will be assigned an individual study ID number, which will be applied to all study documentation.

The study case report form will be used to capture the data required including eligibility, demographics, standard of care treatment data, follow-up data from medical records and routine clinical assessment.

Pseudonymised image data and extracted parameters will be transferred to a dedicated password-protected laptop or PC held by the chief investigator for analysis by the chief investigator or delegated individual.

## Records retention
Archiving for this study will be carried out according to the sponsor's research standard operating procedure, gSOP-17. Destruction of essential documents will require authorisation from the sponsor. Archiving will be authorised by the sponsor following submission of the end-of-study report MRI source data will be stored and archived as per the Trust's standard arrangements for patient's medical records. MRI data and any related electronic and/or paper reports will be stored in line with the Trust's standard arrangements for retention periods for patient medical records and standard UK policy, this is, at least 25 years subject to media being readable. Essential documents will be archived for 5 years after completion of the study.

## Statistical analysis
Using logistic and Cox regression models the study will evaluate:
1. Different thresholds and temporal changes of fMRI parameters (DCE-MRI, DWI-MRI and BOLD-MRI) at various time points during cervical CRT can predict progression-free (local, nodal, distant) survival.
2. Different thresholds and temporal changes of fMRI parameters (DCE-MRI, DWI-MRI and BOLD-MRI) at various time points during cervical CRT can predict response to treatment as defined by TVRR.
3. Different thresholds and temporal changes of fMRI parameters (DCE-MRI, DWI-MRI and BOLD-MRI) at various time points during cervical CRT can predict overall survival.

4. Different radiomics features can predict response to cervical CRT in terms of TVRR, progression-free (local, nodal, distant) survival and overall survival.
5. Different thresholds and temporal changes of fMRI parameters (DCE-MRI, DWI-MRI and BOLD-MRI) at various time points during cervical CRT can predict changes in immunohistochemical markers of hypoxia and epithelial mesenchymal transition.

For each of the above regression analyses, a univariate logistic regression reporting the OR with 95% CI and p value from the likelihood ratio test will be performed. All covariates (fMRI parameters and radiomic features in table 1) and demographics) with a $p<0.1$ will be entered into a multivariate, forward conditional logistic regression and subsequently a $p<0.05$ will be considered statistically significant. The utility of the final prognostic model will be evaluated using sensitivity, specificity and area under the receiver operating characteristic curve.

## Future work
By combining imaging and molecular-based biomarkers, EMPIRIC aims to develop a precise and informative radiogenomic tool, which demonstrates intratumour heterogeneity and in particular regions of radio-resistance within cervical tumours. It will identify relevant parameters and variables from fMRI, biopsy samples and blood samples in the development of predictive and/or prognostic models in cervical cancer radiotherapy. This will allow for further work which will include the development of a larger prospective pilot study aiming to validate this risk stratification model and thereafter a randomised study evaluating the role of personalised dose escalation strategies to fMRI-defined biological targets at the point of brachytherapy delivery based on this risk stratification model.

## Study monitoring plan
The study will be monitored according to the study-specific monitoring plan. Proportionate monitoring will involve completion of remote monitoring forms by the chief/coinvestigator or delegated individual. If the sponsor becomes aware of concerns regarding the project activity at any site, including any deviations from the study protocol, further triggered monitoring may be implemented.

## Ethics and dissemination
The EMPIRIC study received ethical approval from the NHS Health Research Authority (HRA) on 14 February 2022 (protocol number RD2021-29). Confidentiality and data protection measures are strictly adhered to throughout the study. The findings of this study will be disseminated through peer-reviewed publications and scientific conferences, aiming to contribute to the growing body of evidence on the use of multiparametric MRI in cervical cancer management.

**Correction notice** This article has been corrected since it was published. Licence updated to CC BY on 2nd August 2024.

**Acknowledgements** The authors thank the Paul Strickland Scanner Centre for their support, including funding and expertise in scanning patients, image acquisition and analysis.

**Contributors** AL, AP, NJT, HT, YT and PH contributed to study planning, design and protocol development. MA-L, HT, YT and PH contributed to ethical approval. NJT, MA-L, HT, YT and PH contributed to development of MRI aspects. MA-L, HT, YT and PH contributed to development of translational aspects. AL, AP, NJT, HT, YT, PH, MA-L and AC contributed to writing, critical review and revisions.

**Funding** MA-L is supported by the John Bush Charitable Trust (299186). PH is supported by the NIHR Manchester Biomedical Research Centre. This project is supported by the Paul Strickland Scanner Centre.

**Competing interests** None declared.

**Patient and public involvement** Patients and/or the public were involved in the design, or conduct, or reporting, or dissemination plans of this research. Refer to the Methods section for further details.

**Patient consent for publication** Not applicable.

**Provenance and peer review** Not commissioned; externally peer reviewed.

**ORCID iDs**
Mohammed Abdul-Latif http://orcid.org/0000-0001-7206-7210
N Jane Taylor http://orcid.org/0000-0003-0668-9956

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
