## [Reviewer comments · BMJ Open]

ARTICLE DETAILS

TITLE (PROVISIONAL)	An Exploratory study of using Magnetic resonance Prognostic Imaging markers for Radiotherapy In Cervix cancer (EMPIRIC) : A prospective cohort study protocol
AUTHORS	Abdul-Latif, Mohammed; Chowdhury, Amani; Tharmalingam, Hannah; Taylor, N. Jane; Lakhani, Amish; Padhani, Anwar; Hoskin, Peter; Tsang, Yatman

VERSION 1 – REVIEW

REVIEWER	Yusufaly, Tahir Johns Hopkins
REVIEW RETURNED	13-Sep-2023

GENERAL COMMENTS	This looks like a thoroughly planned and well-prepared study. I have no objections at this time.
--

REVIEWER	Felberbaum, RE Universitätsklinikum Ulm Klinik für Frauenheilkunde und Geburtshilfe, Ob/Gyn
REVIEW RETURNED	06-Oct-2023

GENERAL COMMENTS	This manuscript describes the outlines of a study to come. It should be resubmitted after havin obtained results.
---

REVIEWER	Mayrand, Marie-Hélène Universite de Montreal
REVIEW RETURNED	11-Dec-2023

GENERAL COMMENTS	Thank you for giving me the opportunity to review this interesting study protocol. I am confident that I can provide useful feedback on the diagnostic assessment aspects of the protocol. However, other experts in radiotherapy/radiologic imaging will be needed to provide feedback on the technical aspects of the different modalities chosen as interventions. I agree with the authors that improving treatment options for advanced cervical cancer is needed. Functional MRI offers the possibility of identifying “poor responders” to current treatment protocol, with the possibility of testing modified protocols in future trials. I think that at this point, using a single center prospective non-randomized design is appropriate. The inclusion and exclusion criteria should enable the population of interest to be captured. The follow-up schedule, up to two years makes is possible to investigate longer term outcomes.
--

	It is understood that the sample size is determined by the volume of target patients seen at the recruitment center. However, it would add value to the protocol if the authors provided a power calculation. Is it possible that the confidence intervals around the sensitivity and specificity estimates could be so wide as to make reaching any conclusion difficult? In the analysis section, there is mention of a plan to conduct multivariable analysis. The different variables that will be considered for the model should be clearly defined a priori. Given that 14 outcomes are expected, it is unclear how many variables the authors plan to consider for inclusion in their models. In conclusion, this seems like a promising study that could contribute to improve treatment for advanced cervical cancer. A power calculation and some precision around planned analysis would improve the protocol.
--	---

VERSION 1 – AUTHOR RESPONSE

Reviewer: 1

Dr. Tahir Yusufaly, Johns Hopkins

Comments to the Author:

This looks like a thoroughly planned and well-prepared study. I have no objections at this time. We are grateful for your opinion and time to review our study protocol.

Reviewer: 2

Dr. RE Felberbaum, Universitätsklinikum Ulm Klinik für Frauenheilkunde und Geburtshilfe, Klinikum Kempten-Oberallgau GmbH

Comments to the Author:

This manuscript describes the outlines of a study to come. It should be resubmitted after having obtained results.

Thank you for reviewing our manuscript. We would like to highlight that this submission is for a study protocol, not results. We have now reflected this in the manuscript title. We will aim to publish our findings once the study is complete.

Reviewer: 3

Dr. Marie-Hélène Mayrand, Université de Montréal

Comments to the Author:

Thank you for giving me the opportunity to review this interesting study protocol. I am confident that I can provide useful feedback on the diagnostic assessment aspects of the protocol. However, other experts in radiotherapy/radiologic imaging will be needed to provide feedback on the technical aspects of the different modalities chosen as interventions.

I agree with the authors that improving treatment options for advanced cervical cancer is needed. Functional MRI offers the possibility of identifying “poor responders” to current treatment protocol, with the possibility of testing modified protocols in future trials.

I think that at this point, using a single center prospective non-randomized design is appropriate. The inclusion and exclusion criteria should enable the population of interest to be captured.

The follow-up schedule, up to two years makes it possible to investigate longer term outcomes. It is understood that the sample size is determined by the volume of target patients seen at the recruitment center. However, it would add value to the protocol if the authors provided a power

calculation. Is it possible that the confidence intervals around the sensitivity and specificity estimates could be so wide as to make reaching any conclusion difficult?

In the analysis section, there is mention of a plan to conduct multivariable analysis. The different variables that will be considered for the model should be clearly defined a priori. Given that 14 outcomes are expected, it is unclear how many variables the authors plan to consider for inclusion in their models.

In conclusion, this seems like a promising study that could contribute to improve treatment for advanced cervical cancer. A power calculation and some precision around planned analysis would improve the protocol.

Thank you for reviewing our manuscript and your comments.

The following two points have been addressed:

- Power calculation – As this is an exploratory study, sample size calculations have not been performed. Studies of this type are not usually powered to provide estimates of effect size – but instead aim to determine feasibility of different study parameters and provide a guide for sample size estimates in future studies. We have now reflected this in the edited manuscript.
- Definition of variables for multivariate analysis – The study investigates the role of fMRI derived parameters (e.g. ADC, T₂ and R²*) detailed in Table 2 of the manuscript as prognostic and predictive biomarkers in cervical cancer radiotherapy. It is these variables that will form inputs into multivariate analysis. This detail has been made clearer in the manuscript with a reference to Table 2. We hope that these points help address comments by the editor and reviewers.

Kind regards,

Dr. Mohammed Abdul-Latif

VERSION 2 – REVIEW

REVIEWER	Mayrand, Marie-Hélène Université de Montréal
REVIEW RETURNED	07-Feb-2024

GENERAL COMMENTS	Thank you for giving the opportunity to evaluate this protocol a second time. The authors have clarified that this is in fact an “exploratory” study, aimed at planning a bigger “definitive” randomized trial. The authors may want to consider the term “pilot” instead of exploratory. There remain areas in the protocol where it is unclear that this is a pilot, for example in the stated aims: “EMPIRIC is a prospective exploratory cohort study, which aims ...to assess treatment response and predict outcomes in patients; impact: “smooth translation of trial protocol into wider clinical practice”; or chosen outcomes. The protocol would also be improved if there was more specific mention of how exactly the findings will be used to plan a bigger trial. The authors may wish to adjust their timeline, since according to this protocol, recruitment should have been completed in 2023 (“The study aims to recruit 40 patients in total over an accrual period of 24 months at Mount Vernon Cancer Centre, UK. The study started in August 2021 and is actively recruiting.”)
--

VERSION 2 – AUTHOR RESPONSE

Reviewer: 3

Dr. Marie-Hélène Mayrand, Université de Montréal

Comments to the Author:

Thank you for giving the opportunity to evaluate this protocol a second time.

The authors have clarified that this is in fact an “exploratory” study, aimed at planning a bigger “definitive” randomized trial. The authors may want to consider the term “pilot” instead of exploratory.

There remain areas in the protocol where it is unclear that this is a pilot, for example in the stated aims: “EMPIRIC is a prospective exploratory cohort study, which aims ...to assess treatment response and predict outcomes in patients; impact: “smooth translation of trial protocol into wider clinical practice”; or chosen outcomes.

Thank you. Whilst we appreciate the distinction between the terms ‘exploratory’ and ‘pilot’ as study design descriptors, we feel ‘exploratory’ is a more fitting term for this study. Combining multiparametric functional MRI with biopsy and CTC data in a predictive and/or prognostic context is a strategy in its infancy. EMPIRIC will allow for identification of which elements of fMRI, histopathology and CTC are relevant in the generation of predictive and prognostic models before further refining and testing models in pilot study. Nonetheless, we have adjusted the text to make this clearer.

The protocol would also be improved if there was more specific mention of how exactly the findings will be used to plan a bigger trial.

This has now been reflected better in the text (‘Future work’).

The authors may wish to adjust their timeline, since according to this protocol, recruitment should have been completed in 2023 (“ The study aims to recruit 40 patients in total over an accrual period of 24 months at Mount Vernon Cancer Centre, UK. The study started in August 2021 and is actively recruiting. “)

The timeline has been adjusted - we are extending accrual beyond 24 months. Furthermore, the correct year of commencement is 2022 not 2021. This has been corrected.

We hope that these points help address comments by the editor and reviewer.

Kind regards,

Dr. Mohammed Abdul-Latif